# Disturbances of Ocular Circulation in Color Doppler Imaging, Retinal Changes and Electrophysiological Tests with Neuro-Ophthalmological Clinical Symptoms in the Course of CADASIL Syndrome—A Case Report

**DOI:** 10.3390/jcm12051964

**Published:** 2023-03-01

**Authors:** Monika Modrzejewska, Patrycja Woźniak, Wiktoria Bosy-Gąsior, Adam Kaniewski

**Affiliations:** 1II Department of Ophthalmology PUM, Pomeranian Medical University of Szczecin, Al. Powstańców Wielkopolskich 72, 70-111 Szczecin, Poland; 2I Department of Cardiology, Poznan University of Medical Sciences, Długa 1/2 Street, 61-848 Poznań, Poland; 3Scientific Association of Students 2nd Department of Ophthalmology, Pomeranian Medical University in Szczecin, Powstańców Wielkopolskich 72, 70-111 Szczecin, Poland

**Keywords:** CADASIL, small vessel diseases, retinal and choroidal blood flow disturbance, reduced amplitude of the P50 wave in PERG, NOTCH3 gene mutation

## Abstract

The authors present a new paper examining the disturbances in ocular circulation and electrophysiological changes in the presence of neuro-ophthalmic manifestations in a patient with cerebral autosomal dominant arteriopathy with subcortical infracts and leucoencephalopathy (CADASIL). Symptoms reported by the patient included: transient vision loss (TVL), migraines, diplopia, bilateral peripheral visual field loss and convergence insufficiency. CADASIL was confirmed by the presence of *NOTCH3* gene mutation (p.Cys212Gly), the presence of granular osmiophilic material (GOM) in cutaneous vessels in an immunohistochemistry test (IHC) and bilateral focal vasogenic lesions in the white matter of the cerebral hemisphere, with micro-focal infarct in the left external capsule on a magnetic resonance imaging test (MRI). Color Doppler imaging (CDI) confirmed decreased blood flow and increased vascular resistance in the retinal and posterior ciliary arteries, with reduced P50 wave amplitude on a pattern electroretinogram (PERG). An eye fundus examination and fluorescein angiography (FA) revealed the constriction of retinal vessels and a peripheral retinal pigment epithelium (RPE) atrophy with focal drusen. The authors suggest that the cause of TVL may be changes in the hemodynamics of the retinochoroid vessels associated with the narrowing of small vessels and the presence of druses in the retina—which is supported by a decrease in the amplitude of the P50 wave in PERG, changes in OCT correlating simultaneously with changes in MRI imaging and other neurological symptoms.

## 1. Introduction

CADASIL belongs to the group of small vessel diseases (SVD) and is a fairly rare (1.98/100,000) genetic disorder, caused by mutations in the *NOTCH3* gene (on chromosome 19, locus 19p13.1–p13.2; OMIM: 125310), which determines the amino acid sequence of *NOTCH3* protein, playing a crucial role not only in receptor binding but also in intercellular communication [1,2].

The accumulation in the extracellular domain of a receptor protein with smooth muscle cells of small- and medium-sized vessels leads to the formation of 10–15 nm granular osmiophilic material (GOM) deposits and is responsible for the pathophysiology of CADASIL. Furthermore, *NOTCH3* mutation causes explicit changes in the basal membrane and the pericytes of the capillaries. Progressive vessel wall thickening and constriction of the vessels lead to disseminated microangiopathy, followed by the formation of pathological changes mainly in the white matter of the brain but also inside the retina. This process is further intensified by the accumulation of elastic and collagen fibrils in the vessel walls. The alternative hypothesis assumes that GOM deposits are caused by smooth muscle cell degeneration. The similarity in the structure of retinal and cerebral cortical vessels results in the formation of histopathological changes simultaneously in both types of vessels. That is why in such cases MRIs reveal characteristic disseminated focal infarcts. These lesions are most often found in the subcortical and periventricular areas of the brain’s white matter, mainly in the temporal lobe, external capsule, the frontal and parietal lobe, subcortical nuclei and in the centrum semiovale. Characteristic lesions are also located in the grey matter of the brain and result in neuronal apoptosis, particularly in the third and the fifth ventricles. A gradual onset of CADASIL symptoms occurs between the ages of 20 and 30, and most often manifests as subcortical infarcts, transient ischemic attack (TIA), transient vision loss (TVL), episodes of migraine (often with aura), apathy, dementia and other psychiatric disorders followed by acute encephalopathy with mood disturbances [3,4,5,6]. Those diagnosed with CADASIL can expect to be bedridden around 60 years of age after suffering repeated strokes. The life expectancy for men is around 65 years of age and for women it is around 70 years of age.

According to the literature, the most common manifestations of CADASIL syndrome include neurological symptoms, such as TIA as well as of ocular origin as TVL, the basis of which may be small vascular changes and blood flow disorders in both the cerebral and retinal vessels. We could not find many studies on ophthalmic manifestations in patients with CADASIL (keywords: “ophthalmic symptoms” and “CADASIL” PubMed: 34 articles, and “neuro-ophthalmological symptoms” and “CADASIL” Google Scholar: 45 articles). Only one of them (between PubMed and Google Scholar) focused solely on retinal flow disturbances, assessed by a different method than in our study, scanning laser Doppler flowmetry.

Another unreported phenomenon is hemodynamic disturbances in the optic nerve head and retrobulbar vessels, both in the central retinal artery and in the choroidal arteries, in the posterior ciliary arteries measured using CDI (color Doppler imaging). Moreover, only two publications discuss the interpretation of the results of electrophysiological studies (pERG) [7,8]. Therefore, it seems important to present original ophthalmological observations combined with an analysis of their relationship with neuro-ophthalmic clinical symptoms in the course of CADASIL, considering their vascular origin.

## 2. Case Description

A 43-year-old woman was referred for ophthalmological consultation due to a positive history of transient vision loss, diplopa, near heterophoria, defects in peripheral kinetic perimetry and problems with accommodation with convergence. The patient’s medical history showed CADASIL disease with typical neurological (severe migraines, transient aphasia, loss of memory, skin sensation disorder and numbness of the right-upper limb) and cardiological symptoms, including tachycardia and premature supraventricular contractions (PVCs). A neuropsychological examination did not reveal any cognitive function disorders [9]. CADASIL was also confirmed in the patient’s mother, whose symptoms were predominantly dementia and cognitive impairment, despite having the same gene sequence (*NOTCH3* gene on chromosome 19—locus 19p13.1–p13.2; OMIM: 125310).

The acuity of vision of both eyes was not disturbed once the episodes of transient vision loss had subsided (Snellen chart 1.0; no retractive disorders: the right eye +0.25/−0.5/93, the left eye +2.0/−0.5/45). Concomitant symptoms included loss of the nasolabial fold as well as increased deep tendon reflexes of the upper-left limb. The orthoptic examination revealed convergence insufficiency (Figure 1) and left-eye exophoria, confirmed with the Maddox test; however, neither the assessment of binocular vision nor the examination of the mobility of the extraocular muscles in the Hess screening revealed any dysfunction.

The abnormalities we confirmed were as follows: bilateral peripheral narrowing of the visual field up to 20° in both eyes, with a defect in the lower part of the left eye’s visual field in kinetic perimetry examination.

The intraocular pressure (ICARE-tonometr) was within the normal range (20.6 mmHg in both eyes). No abnormalities appeared on the optical coherent tomography (OCT) of the optic nerve, the ganglion cell complex (GCL + IPL) or the static perimetry. The macular OCT test did, however, show thickening of the central part of the macula (Figure 2), but macular angio-OCT and visual evoked potential (VEP) tests showed no anomalies. Indirect ophthalmoscopy revealed the constriction of retinal arteries, arterial wall thickening and arteriovenous nicking (Figure 3A). In addition, there were other interesting findings, namely peripheral RPE atrophy and focal hyperfluorescence in the temporal macula region at the inferior vascular temporal arch (Figure 3B,C), which was thereafter confirmed in the fluorescein angiography (FA) (Figure 4) and OCT of the left eye.

The retinal drusen were observed in the retina of the left eye (Figure 5).

Late-phase indocyanine green angiography (ICGA) also confirmed the presence of lesions, demonstrated by the small, linear, focal hyperfluorescence of the lower part of the central fovea exclusively in the left eye. It is not certain whether these changes are the result of focal RPE atrophic lesions.

Color Doppler imaging (CDI) confirmed decreased blood flow in both the retinal and choroid arteries (central retinal artery and posterior ciliary arteries), which was clearly marked in the left eye, as well as increased vascular resistance in these arteries (Table 1).

A pattern electrophysiological test (PERG) demonstrated reduced P50 wave amplitude in the patient’s left eye, while a visual evoked potential (VEP) test did not reveal any abnormalities (Figure 6).

An MRI of the central nervous system showed multiple focal vasogenic lesions in the white matter of both hemispheres, which were partly merged in the occipital horn region (Figure 7B). A single microinfarct lesion was found in the left external capsule (Figure 7A).

An ultrastructural examination of endothelial changes in both muscle and skin biopsies revealed the presence of GOM in small vessels of the skin and the muscle (Figure 8).

The examination also demonstrated discrete degenerative lesions of the pericytes and myocytes of the vessels, along with vacuolation and constriction. Finally, genetic testing confirmed *NOTCH3* gene mutation (p. Cys212Gly allele; OMIM: 125310). The same diagnosis was confirmed in the patient’s mother, who has also manifested CADASIL symptoms, including dementia and muscle and neurological dysfunction with confirmed *NOTCH3* gene mutation (p. Cys212Gly allele; OMIM: 125310).

## 3. Discussion

The first reports of CADASIL were described as familial Binswanger disease by Van Bogaert in 1955 [10]. The first symptoms of this affliction were noted in 1977, but it was not until 1993 that a complete description of the disease, including co-ocurrence of ischemic infarcts, progressive dementia and migraines, was published by Turnier-Lasserve et al. [4]. Because histopathological lesions were only found in the arteries, the disease was described as autosomal dominant arteriopathy with subcortical infracts and neurodegeneration of the white matter in the brain. In 1996, it was confirmed that the *NOTCH3* gene, located on chromosome 19 (p13.1–13.2), is responsible for the onset of symptoms.

CADASIL steadily worsens the everyday functioning of patients because of its numerous neurological implications, which are caused by vasogenic defects. The source of this condition is thought to be GOM deposits, which are found not only in the brain but also in skeletal muscles, kidneys, pericardium, skin and retinas [11].

The retinal changes in CADASIL include retinal arteriolar constriction, decreased A/V ratio, arteriovenous nicking, artery straightening and enhanced vascular reflex [12], all of which were manifested by the patient in this clinical presentation. Some authors emphasize the importance of peripheral RPE atrophic lesions along with focal drusen, which have not been widely described before [12,12,13]. The foci of grouped single drusen formations were observed in our patient along the retinal vessels, and it is possible they might be related to GOM deposits. In the available literature, other retinal manifestations of CADASIL include the reduction in optic nerve fiber layer thickness [14,15] as well as reduced vessel density in the deep retinal plexus [16]. Another new observation, not yet published, is the reduction in retrobulbar blood flow and increased vascular resistance of the central retinal artery and posterior ciliary arteries, found symmetrically in the CDI but mainly in the left eye of the patient (Table 1). In 2004, Harju et al. measured decreased capillary retinal blood flow using another research method (scanning laser flowmetry), in a group of patients with CADASIL but without visual impairment. The measurement concerned peripapillary retinal circulation 1 and 2 mm from the disc rim and did not include patients with an attack of TIA or TVL [17].

Our observations of the changes in the ocular circulation in the small vessels of the retina and choroid in color Doppler imaging may be related to the hemodynamic disturbances also suggested but not confirmed by Pretegiani et al. [12] and with the findings of Neils et al. [16], who described changes in angio-OCT that may account for capillary plexus thinning as a result of pericyte dysfunction in retinal capillaries associated with the *NOTCH3* mutation. Additionally, recent observations by Lin et al. suggest that reduced macular vessel density and inner retinal thickness correlate with the severity of cerebral autosomal dominant arteriopathy with subcortical infarcts and leukoencephalopathy (CADASIL), which may also be of great importance in the future use of the CDI method for diagnosing the advancement of this disease [18].

It is possible that in the future a decrease in blood flow velocity and increase in RI in chorioretinal capillaris detected using CDI and confirmed using OCT-A may serve as an additional disease marker in CADASIL and other small vessel diseases.

Another interesting observation concerns the early changes in functional retinal ganglion cells and retinal photoreceptors, which were confirmed in the pattern electroretinogram (PERG) and revealed a reduced P50 wave amplitude in the patient’s left eye (Figure 6). A similar phenomenon has been described as cholinergic deficiency in the pathogenesis of Alzheimer’s disease (AD); therefore, it is possible that it is related to dementia changes [19]. The PERG revealed a retinal bioelectric dysfunction in the form of reduced P50 wave amplitude in the patient’s left eye, which was also confirmed in our patient. These changes may indicate possible ischemic impairment of ganglion cells and their axons as a result of retinal photoreceptor dysfunction, as documented in our CDI study with concomitant changes in macular OCT [20].

Other retinal lesions described in the literature include cotton wool spots, which are a result of the obliteration of the arterioles, caused by deposits of axoplasm in retinal nerves. The correlation between retinal effusion and lacunar infarcts in patients with SVD has also been proven, but was not observed in the reported case [21,22].

The co-occurrence of diplopia and transient vision loss requires excluding vertebral artery insufficiency with the disturbance of blood flow in the optic nerve head [23], in addition to migraine episodes with aura, scotoma, or photopsia and blurred vision [12]. Non-arteritic anterior ischemic optic neuropathy is described as the cause of total vision loss [24].

Diplopia is a rare symptom mentioned in CADASIL and, according to Davous et al., may be triggered by vasogenic lesions in the brain stem, which are caused by vertebrobasilar insufficiency, tumor, or stroke of the occipital region [25,26,27].

In this clinical report, convergence insufficiency might have been the result of impaired extraocular muscle innervation (III-cranial nerve) or exophoria (latent divergent strabismus); other literature mentions the possibility of: ophthalmoplegia, Perinaud syndrome, myasthenia, oculomotor nerve palsy or thyroid-related orbitopathy [28].

In the course of CADASIL, strokes are characterized as focal regions of necrosis (sinus or lacunar infarcts) and are mainly located in the white matter of the brain, particularly around the frontal lobe, temporal lobe and cortical nuclei. They are rarely located in the spinal cord or brain stem region, although such localization was shown in our patient’s history. An infarct in the temporal lobe can often lead to the disturbance of visual spatial orientation, narrowing interests, apathy, behavioral changes and memory and concentration disturbance, all of which were manifested by our patient.

Further examples of symptoms include frequent migraines with visual or sensory aura, mostly manifesting between the third and fourth decade of life and usually preceding the identification of morphological vascular lesions [3], as in this case.

The clinical presentation of CADASIL may differ, even among family members who are carriers of the same gene mutation, and is characterized by symptoms including migraine with aura, subcortical ischemic events, mood disturbances, apathy and cognitive impairment [29].

Ultrastructural examination of both muscle and skin biopsies, revealing the presence of GOM in small vessels of the skin and the muscle, is a strong indicator of CADASIL, which can be further confirmed by the presence of degenerative lesions of the pericytes and myocytes of the vessels, along with vacuolation and constriction [30]. The genetic testing for *NOTCH3* gene mutation (p. Cys212Gly allele; OMIM: 125310) provides final confirmation of the disease [31]. Such a diagnosis was made in the case of our patient.

The causal treatment includes gene therapy and inhibition of culprit protein translocation. New in vitro and in vivo trials have been carried out and may provide new therapeutic possibilities for patients suffering from genetic diseases. Researchers also hope to succeed in using immunotherapy [31,32], growth factors [33,34] and exon skipping [35]. Initial results are promising and these have already been implemented as groundbreaking therapeutic options for patients suffering from other neurological diseases. For instance, antisense oligonucleotide-induced exon-skipping enables synthesis of partially functional dystrophin in patients with Duchenne muscular dystrophy [36]. Another successful example is the alternative splicing of SMN2 protein pre-MRA, which consecutively leads to the higher expression of functional SMN protein in patients with spinal muscular atrophy (SMA) [37]. Treatment of CADASIL with the antiplatelet agents used to prevent stroke recurrence in conventional stroke has not been proven to be effective. There are also no drugs that show an effect against dementia, so no treatment has been established.

New treatment options have already been used in patients suffering from other neurological diseases [36,37], although the side effects of such therapies still limit their use on a larger scale. That is why it is crucial to find *NOTCH3* gene expression variants that will provide successful therapeutic effects with minimal side effects.

In differential diagnosis, other maculopathies, such as Stargardt macular dystrophy or macular edema, should also be taken into consideration [37]. Among patients suffering from AD, the correlation between PERG abnormalities and the reduction in optic nerve fiber layer thickness was confirmed using OCT [7].

## 4. Conclusions

Clinical symptoms in the form of transient vision loss, diplopia, convergence disorders, migraine episodes (with aura) together with ischemic stroke and premature dementia in young and middle-aged patients, without the presence of risk factors for vascular disease, can be considered in the diagnosis of CADASIL. Additional ocular symptoms are RPE retinal atrophy, retinal vasoconstriction, focal drusen, and decreased blood flow in the chorio-retinal region with pERG reduction in P50, possibly as a result of a cholinergic deficit or possible dysfunction of macular or ganglion cells. Hyperintensive changes in the white matter of the brain in an MRI, GOM deposits in small vessels of the skin and muscles revealed in ultrastructural examination of muscle and skin biopsies, as well as confirmed mutation of the *NOTCH3* gene are particularly important in the CADASIL diagnostic process [38]. Circulatory disorders in the choroidal and retinal capillaries in CDI and confirmed using OCT-A could offer an additional early diagnostic marker in CADASIL or other small vessel diseases.

## Figures and Tables

**Figure 1 jcm-12-01964-f001:**
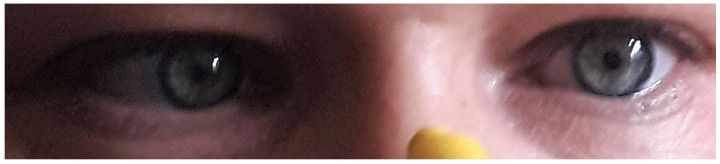
Convergence insufficiency of the left eye. The patient has agreed to the publication of this photo—proper consent was obtained.

**Figure 2 jcm-12-01964-f002:**
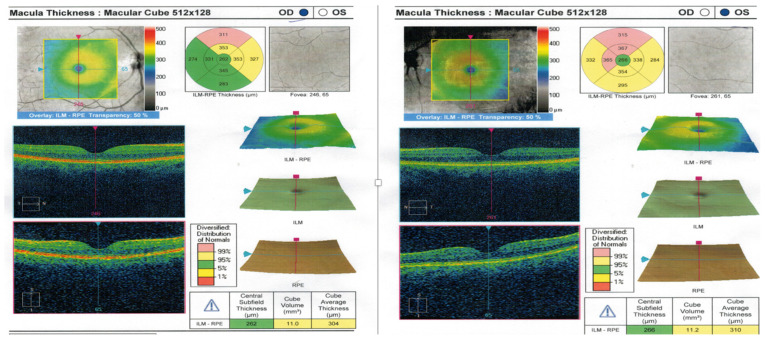
Thickening of the central part of the macula demonstrated in macular OCT, mainly in the left eye.

**Figure 3 jcm-12-01964-f003:**
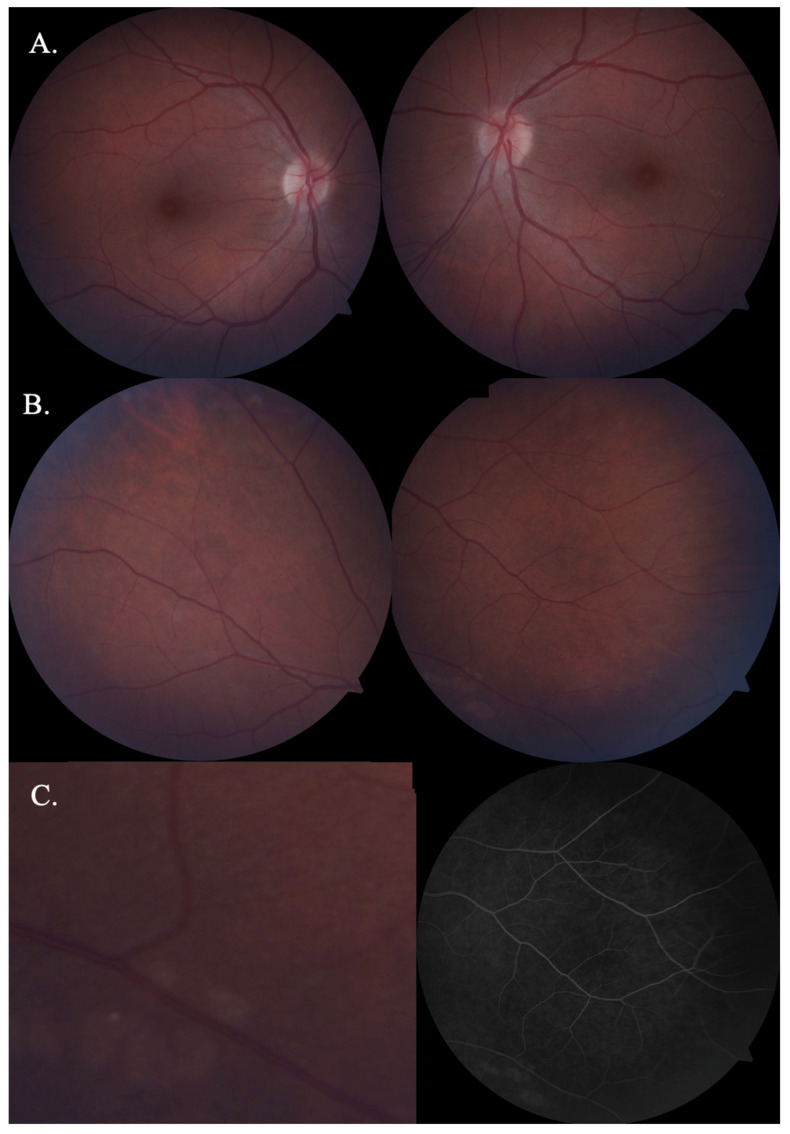
Eye fundus fluorescein angiography of the patient with CADASIL. (**A**) Constricted retinal arteries in both eyes. (**B**) Retinal pigment epithelium (RPE) atrophy in both eyes, but mainly in the left eye. (**C**) Retinal pigment epithelium (RPE) atrophy with focal drusen in the left eye.

**Figure 4 jcm-12-01964-f004:**
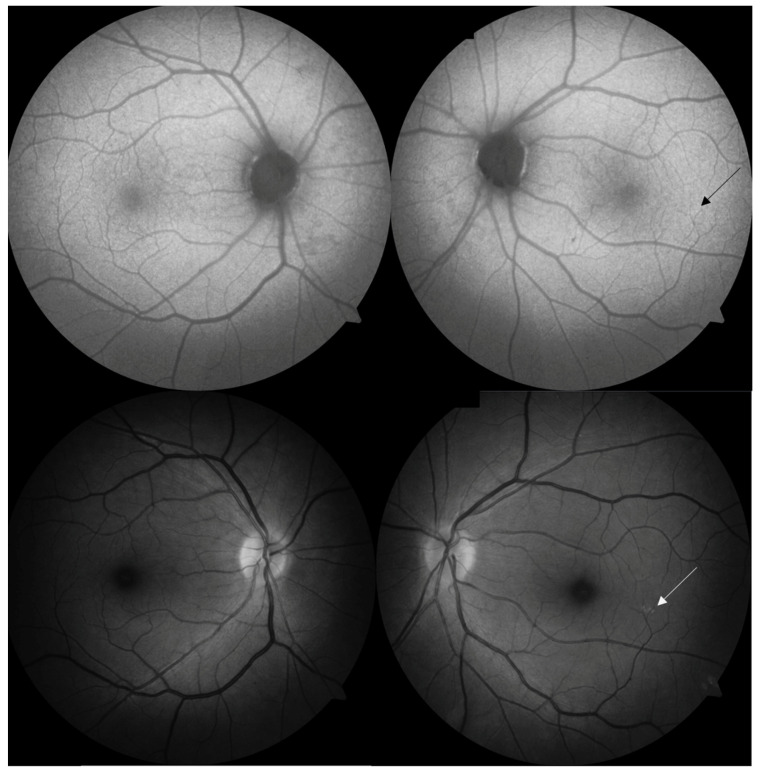
Fluorescein angiography of the right and the left eye. Discrete retinal lesions in the left eye (the arrow demonstrates focal drusen).

**Figure 5 jcm-12-01964-f005:**
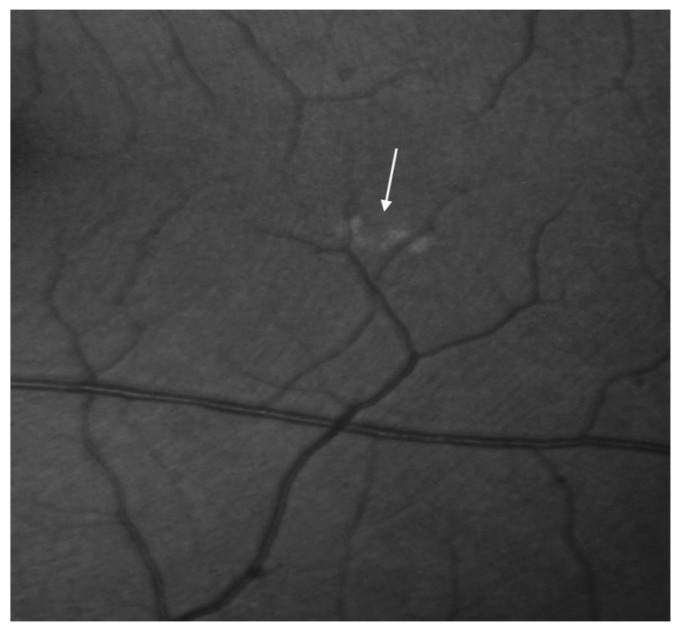
Focal drusen around the subretinal region of the left eye demonstrated in the fluorescein angiography (the arrow demonstrates focal drusen).

**Figure 6 jcm-12-01964-f006:**
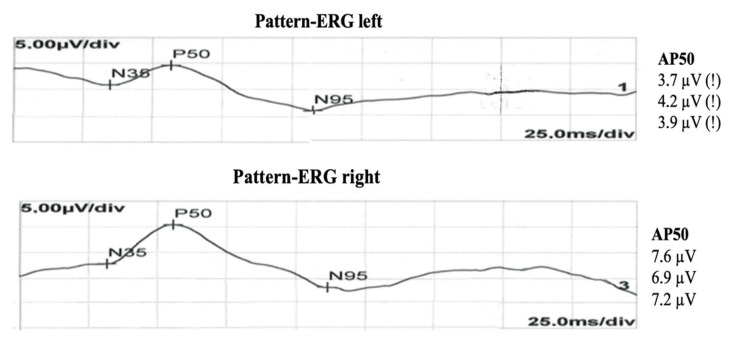
Pattern electroretinogram demonstrating the reduced P50 wave amplitude in the patient’s left eye. No abnormalities in the right eye.

**Figure 7 jcm-12-01964-f007:**
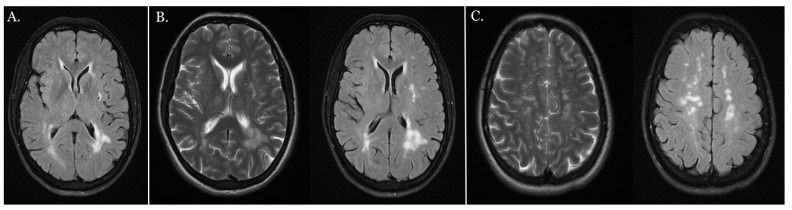
MRI of the central nervous system. (**A**) Single focal subcortical microinfarct in the left external capsule shown in MRI-FLAIR image. (**B**) Subcortical and periventricular areas of the white matter of the brain shown using MRI, along with multiple focal vasogenic lesions in the white matter of both hemispheres, partly merged in the occipital horn region (LS: MRI-T2 image, RS: MRI-FLAIR image). (**C**) The formation of pathological changes mainly in the white matter of the brain shown in MRI (left side (LS): MRI-T2 image, right side (RS) MRI-FLAIR image).

**Figure 8 jcm-12-01964-f008:**
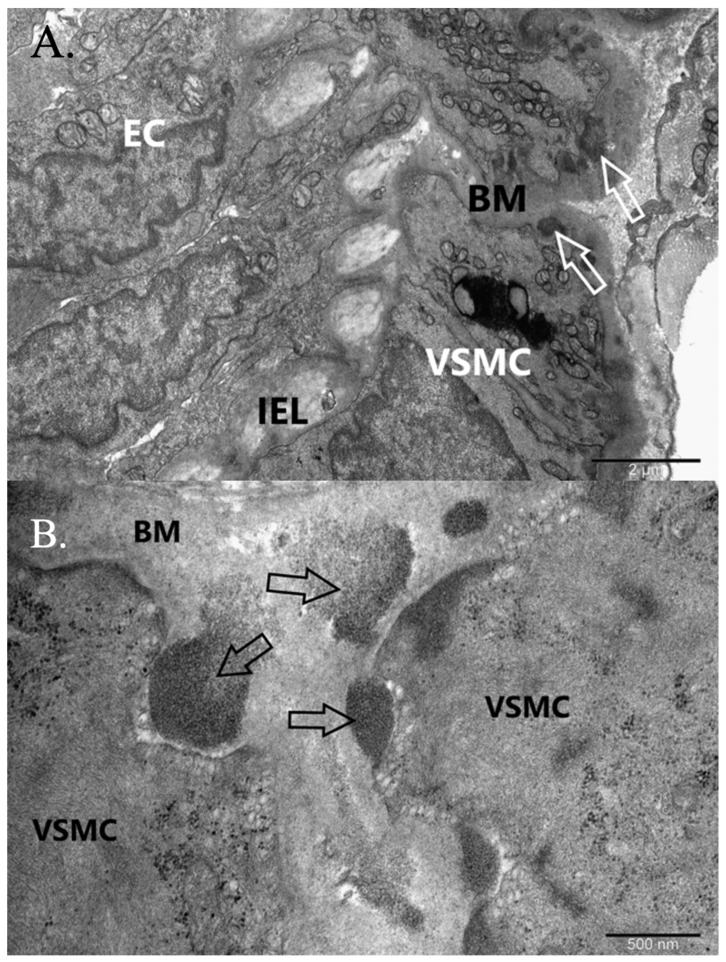
Granular osmiophilic material (GOM) in cutaneous small vessel wall in immunohistochemistry laboratory test (IHC). (**A**) Cross-section of an arteriole with GOMs. VSMC, vascular smooth muscle cell; EC, endothelial cell; BM, basement membrane; IEL, internal elastic lamina; arrow, GOM. Orig. magn. ×15,000. (**B**) GOMs magnified. VSMC, vascular smooth muscle cell; BM, basement membrane; arrow, GOM. Orig. magn. ×50,000.

**Table 1 jcm-12-01964-t001:** Color Doppler imaging with decreasing blood flow in central retinal artery and posterior ciliary arteries.

	Vs RE cm/s	Vs LE cm/s	Vd RE cm/s	Vd LE cm/s	RI RE	RI LE	PI RE	PI LE
OA	58.59	56.6	23.24	21.45	0.61	0.62	0.94	0.99
CRA	11.72	8.06	2.99	1.17	0.71	0.9	1.21	1.6
PCA medial	17.04	10.74	5.99	3.12	0.65	0.71	0.98	1.21
PCA lateral	20.73	12.21	3.45	3.22	0.71	0.83	0.89	1.35

Abbreviations: OA, ophthalmic artery; CRA, central retinal artery; PCA, posterior ciliary artery; RI, resistivity index; PI, pulsatility index; RE, right eye; LE, left eye; Vs, systolic velocity; Vd, diastolic velocity.

## Data Availability

The data presented in this study are available on request from the corresponding author. The data are not publicly available due to privacy reasons.

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
