# Peer review of "Disturbances of Ocular Circulation in Color Doppler Imaging, Retinal Changes and Electrophysiological Tests with Neuro-Ophthalmological Clinical Symptoms in the Course of CADASIL Syndrome—A Case Report"

_jcm, 2023, doi:10.3390/jcm12051964_

Round 1
Reviewer 1 Report
INTRODUCTION some words are unnecessarily repeated in the introductory part like…transient vision loss…
PAGE 2 LINE 64 …ophthalmic…missing word
“ a 43-year-old woman was referred for ophthalmological consultation due to a positive history of several microinfarcts, diplopia, transient vision loss, problems with accommodation and convergence, severe migraines, transient aphasia, loss of memory, skin sensation disorder and numbness of the right upper limb” leave out neurological symptoms, as patients are not referred to an ophthalmologist to solve the problem of aphasia and memory loss.
State the type of visual field test that has been done in this case and on which perimeter. Describe In more detail visual field defects.
“The intraocular pressure (ICARE-tonometr) was within the normal range (20.6 97 mmHg). There were no abnormalities shown in the optical coherent tomography (OCT) 98 of the optic nerve, ganglion cell complex (GCL+IPL), or in static perimetry.” The authors stated earlier in the text that there were visual field defects.
Author Response
Responses to Reviewer 1
Thank you for your interest in our article. Thank you for a thorough and insightful look at the article which seems to be interesting and at the same time it is an additional opportunity to analyze or differentiate the type of disturbances in a CADASIL patient - a topic that has not been noticed and described so far. The author of this article is an experienced Doppler sonographer, hence his interest in the patient's case.
- INTRODUCTION some words are unnecessarily repeated in the introductory part like…transient vision loss…
Authors: Transient ischemic attack (TIA), transient vision loss (TVL) both expressions are ambiguous, therefore they have been included in the article, TIA is of brain origin, while TVL - ophthalmologic, therefore, the authors consider the two definitions to be appropriate - in addition to changes in the brain, the patient also developed haemodynamic disturbances in postbulbar vessels.
- PAGE 2 LINE 64 …ophthalmic…missing word
Authors: Sentence corrected and supplemented.
- “ a 43-year-old woman was referred for ophthalmological consultation due to a positive history of several microinfarcts, diplopia, transient vision loss, problems with accommodation and convergence, severe migraines, transient aphasia, loss of memory, skin sensation disorder and numbness of the right upper limb” leave out neurological symptoms, as patients are not referred to an ophthalmologist to solve the problem of aphasia and memory loss.
Authors: Thank you for your correct comment - the sentence has been changed and we refer only to ophthalmologic symptoms. According to the reviewer's comment: neurological symptoms have been thrown out in this sentence.
- State the type of visual field test that has been done in this case and on which perimeter. Describe in more detail visual field defects.
Authors: There are two types of perimetry, i.e. field of vision tests - kinetic and static fields - kinetic tests the entire the whole of field of view, mainly peripheral - static only limited to the field in the range of 24-30 degrees from the fixation point; our changes describe the changes in the large peripheral field that were noted, while the perimeter of the small very limited field did not register changes because they were not there.
- “The intraocular pressure (ICARE-tonometr) was within the normal range (20.6 97 mmHg). There were no abnormalities shown in the optical coherent tomography (OCT) 98 of the optic nerve, ganglion cell complex (GCL+IPL), or in static perimetry.” The authors stated earlier in the text that there were visual field defects.
Authors: The authors describe only changes in kinetic perimetry, but there were no changes in static perimetry describing changes in field sensitivity up to 30 degrees from the fixation point. The opinion is consistent.
The authors of the article would like to thank the reviewers for their helpful comments which
raised the rank of the text and made the article interesting for the recipient-reader.
Best regards,
Monika Modrzejewska

Reviewer 2 Report
This full paper on the CADASIL case is a case report, but includes detailed ophthalmologic macular findings and retinal potential maps as well as brain MRI and muscle pathology findings. Genetic testing has also demonstrated a NOTCH3 mutation. Overall, it is not only very interesting but also scientifically credible. The only point to consider, which is not a negative point, is that it is a single case report: what should the journal make of a single case report...? However, as a reviewer I would rate this case report as a very significant study, equivalent to an original article. In my opinion, for this one case report, a more detailed and accurate description of the findings, not just the ophthalmologic findings, is needed.
Minor1/ L40: The NOTCH3 gene in CADASIL is 125310 in OMIM.
MInor2/ L44: Please spell out the abbreviation "GOM". (GOM are granules that stain darkly osmium under an electron microscope)
Minor3/ L59: The first symptom of CADASIL is a migraine attack with aura, often occurring around the age of 20 to 30 years.
Minor4/ L62: The prognosis for CADASIL is bedridden at around 60 years after repeated stroke. Men die at around 65 years of age and women at around 70 years of age.
Minor5/ L62: The treatment of CADASIL has not been proven effective with antiplatelet agents, which are used to prevent recurrent strokes in conventional stroke. There are also no drugs that have been shown to have an anti-dementia effect, so no treatment has been established. It is important to add this statement.
Minor6/ L69: Regarding this sentence, what type of search engine did you search and systematically research?
MInor7/ L140: The left eye of the ERG in the pattern is low amplitude. Were there any salt-and-pepper, cherry spots or other findings specific to CDASIL in the retinal findings?
Minor8/ L142-145: Brain MRIs are usually aligned from the base of the brain towards the parietal region. Therefore, how about the images in Figure 7 as C → B → A from left to right?
Minor9/ L145: Pineal cysts are not evident from these MRI slices; they are less relevant to CADASIL, so an MRI could be added or sentences could be deleted.
Minor10/ L147: Please describe the MRI imaging: is it a T2-weighted image or a FLAIR image?
Minor11/ L155: This is a very good IHC photograph. Is the kit used for this immunostaining the Avicin-Biosin complex kit? Please add a note if you know.
Minor12/ L162: The NOTCH3 notation is genetic, so why not use diagonal italics?
Best regards.
Dr. Reviewer
Author Response
Responses to Reviewer 2
Thank you for your interest in our article. Thank you for a thorough and insightful look at the article which seems to be interesting and at the same time it is an additional opportunity to analyze or differentiate the type of disturbances in a CADASIL patient - a topic that has not been noticed and described so far. The author of this article is an experienced Doppler sonographer, hence his interest in the patient's case.
This full paper on the CADASIL case is a case report, but includes detailed ophthalmologic macular findings and retinal potential maps as well as brain MRI and muscle pathology findings. Genetic testing has also demonstrated a NOTCH3 mutation. Overall, it is not only very interesting but also scientifically credible. The only point to consider, which is not a negative point, is that it is a single case report: what should the journal make of a single case report...? However, as a reviewer I would rate this case report as a very significant study, equivalent to an original article. In my opinion, for this one case report, a more detailed and accurate description of the findings, not just the ophthalmologic findings, is needed.
Authors: Due to the fact that the patient is under constant neurological care and has a full diagnosis of Cadasil disease - the intention of the authors was to thoroughly assess the patient's ophthalmological condition due to the ocular symptoms reported by him, including blindness - which may also be of ocular origin. We assume that since the case is well documented genetically, neurologically and ultrastructurally, there is no need to present the neurological findings more extensively. We focused on those elements that help in differentiation from an ophthalmological point of view - and are not described in the literature, such as color Doppler imaging of the retrobulbar arteries, assessment of changes in the hemodynamics of the optic nerve head and in the central retinal artery with in the posterior ciliary arteries, and the coexisting functional change of photoreceptor cells of the retina around the macula, in which vascular changes visible in OCT of the macula were also confirmed.
--------------------------------------------------------------------------------
- Minor1/ L40: The NOTCH3 gene in CADASIL is 125310 in OMIM.
Authors: All changes have been made.
2. Minor2/ L44: Please spell out the abbreviation "GOM". (GOM are granules that stain darkly osmium under an electron microscope) Authors: Abbreviation has been clarified and added to the text.
- Minor3/ L59: The first symptom of CADASIL is a migraine attack with aura, often occurring around the age of 20 to 30 years.
Authors: There were added to the text.
- Minor4/ L62: The prognosis for CADASIL is bedridden at around 60 years after repeated stroke. Men die at around 65 years of age and women at around 70 years of age.
Authors: There were added to the text.
- Minor5/ L62: The treatment of CADASIL has not been proven effective with antiplatelet agents, which are used to prevent recurrent strokes in conventional stroke. There are also no drugs that have been shown to have an anti-dementia effect, so no treatment has been established. It is important to add this statement.
Authors: A fragment of text has been added to the article according to reviewers suggestion. - Minor6/ L69: Regarding this sentence, what type of search engine did you search and systematically research?
Authors: Added types of search engines to the introduction - and slightly changed the essence of this fragment of text.
We have found not many studies about ophthalmological manifestations in CADASIL disease (number of articles was listed in the manuscript). Only 1 of them (Pubmed and Google Scholar) focused only on retinal flow disturbances, but assessed by a different method (than in our study) such as Scanning Laser Doppler Flowmetry. Our manuscript was supplemented with this study after receiving peer review.
- Minor7/ L140: The left eye of the ERG in the pattern is low amplitude. Were there any salt-and-pepper, cherry spots or other findings specific to CADASIL in the retinal findings?
Authors: There were single and group teams that were seen in volleyball that were photographed and presented in the pictures,indicating with arrowheads. Besides, in the ophthalmic image, the vessels of the choroid that indirectly in the Doppler examination, they indicate a disturbance and change in blood flow velocity, which could also indicate functional disorders of the retina and changes in the ERG test.
- Minor8/ L142-145: Brain MRIs are usually aligned from the base of the brain towards the parietal region. Therefore, how about the images in Figure 7 as C → B → A from left to right?
Authors: The order of the photos has been changed, saved according to reviewer's suggestion.
- Minor9/ L145: Pineal cysts are not evident from these MRI slices; they are less relevant to CADASIL, so an MRI could be added or sentences could be deleted.
Authors: At the suggestion of the reviewer, the mention of changes in the pineal gland has been removed.
- Minor10/ L147: Please describe the MRI imaging: is it a T2-weighted image or a FLAIR image?
Authors: The information was supplemented.
- Minor11/ L155: This is a very good IHC photograph. Is the kit used for this immunostaining the Avicin-Biosin complex kit? Please add a note if you know.
Authors: We obtained a photo from the patient's archive - we do not know the staining used in this photo, so we cannot extend this description.
- Minor12/ L162: The NOTCH3 notation is genetic, so why not use diagonal italics?
Authors: Oblique italics are used for the NOTCH3 notation.
The authors of the article would like to thank the reviewers for their helpful comments which
raised the rank of the text and made the article interesting for the recipient-reader.
Best regards,
Monika Modrzejewska

Round 2
Reviewer 1 Report
Dear authors,
Thank you for accepting my suggestions.
Best regards.